# Physical Activity and Psychosocial Outcomes in Adults with Achondroplasia: An Exploratory Study

**DOI:** 10.3390/ijerph21091160

**Published:** 2024-08-31

**Authors:** Inês Alves, Orlando Fernandes, Maria António Castro, Sofia Tavares

**Affiliations:** 1School of Health and Human Development, University of Évora, Comprehensive Health Research Centre, CHRC, 7002-554 Évora, Portugal; orlandoj@uevora.pt; 2National Association for Skeletal Dysplasias, ANDO Portugal, 7005-144 Évora, Portugal; 3Higher School of Health, CitechCare, CDRSP, Polytechnic Institute of Leiria, 2411-901 Leiria, Portugal; maria.castro@ipleiria.pt; 4Department of Psychology, Center for Research in Education and Psychology of the University of Évora (CIEP-UE), University of Évora, 7000-727 Évora, Portugal; tavares.sofia@uevora.pt

**Keywords:** skeletal dysplasia, rare disease, exercise, wellbeing

## Abstract

Background: Adults with achondroplasia face physical and psychosocial challenges that may impact their health-related quality of life and mental health. This exploratory cross-sectional study aimed to investigate relationships between health-related quality of life, mental health, and physical activity levels in adults with achondroplasia, focusing on potential gender differences. Methods: Sixteen adults with achondroplasia (10 women, 6 men; age 37.2 ± 13.5 years) completed the Short Form Health survey, the Brief Symptom Inventory, and the International Physical Activity questionnaire. Descriptive statistics, non-parametric group comparisons, correlational analyses, and linear regressions were conducted. Results: Moderate physical activity showed strong positive correlations with general health (rs = 0.79, 95% CI [0.50, 0.92]), vitality (rs = 0.60, 95% CI [0.15, 0.85]), and physical functioning (rs = 0.62, 95% CI [0.18, 0.86]), on SF-36. Women reported lower quality of life scores than men across most SF-36 dimensions. Significant gender difference was observed in vitality (r = 0.61) and pain (r = 0.55). Physically active participants presented better outcomes in general health (r = 0.63) and vitality (r = 0.55) compared to inactive participants. Conclusions: This study provides preliminary evidence suggesting potential benefits of moderate-intensity physical activity on health-related quality of life and mental health among adults with achondroplasia, with notable gender differences. While limited by sample size and study design, the findings highlight the need for larger, longitudinal studies to further explore the role of physical activity in enhancing well-being in this population.

## 1. Introduction

Achondroplasia (ACH), a rare skeletal dysplasia caused by mutations in the fibroblast growth factor receptor 3 (FGFR3) gene with a prevalence of 1 in 25,000, is observed in both males and females equally [1]. It results in disproportionately short stature, six standard deviations below the normal population [2].

Beyond physical impairments, adults with achondroplasia face considerable psychological and psychosocial challenges stemming from social stigma and difficulties in daily living participation, and the unique experience of navigating a world designed for average-sized individuals [3]. As individuals with achondroplasia reach adulthood, they encounter intensified negative stereotypes related to short stature, potentially impacting their perceived competence and suitability for certain occupations [3,4]. While research on psychosocial well-being in this population has been extremely limited, the existing literature reveals a concerning picture. Studies have documented restrictions in social participation [5], reduced material well-being [6], and high rates of depression, anxiety, and pain in this population [7]. Previous studies examining health-related quality of life (HrQoL) in adults with achondroplasia have yielded mixed results. Some research has indicated that, mostly due to the demands of dealing with the numerous social and physical disadvantages related to their condition [5,8], adults with achondroplasia suffer significantly lower physical and mental well-being and three times the rate of psychiatric illness diagnosis as compared to the general population [4]. Others have indicated that the functional health status of adults with achondroplasia is not drastically reduced in comparison with that of the general population [6]. These inconsistencies underscore the need for further investigation into the factors influencing HrQoL in this population.

Physical symptoms experienced by individuals with achondroplasia, such as chronic pain and fatigue, may adversely affect their HrQoL [4,7,9]. Pfeiffer et al. (2022) indicated that individuals with achondroplasia presented difficulty walking long distances due to having low stamina or tiring easily. Additionally, psychosocial factors like lack of social support have been associated with lower mental health scores [5]. Mental health issues include concerns about declining health and function with aging, as well as challenges with interpersonal relationships [10]. Self-esteem plays a crucial role in shaping and reflecting the quality of life experienced by individuals with disabilities, particularly in relation to self-adaptation to the disability [11]. However, adults with achondroplasia have reported significantly lower self-esteem, psychological disorders resulting from inferiority complex, and dissatisfaction with physical appearance [5]. Women have shown more problems with self-esteem, body image, and acceptance difficulties in comparison to men [12].

Positive associations between physical activity and well-being in the general population are well-established [13,14], yet despite emerging evidence of mental health issues and psychosocial participation restrictions in adults with achondroplasia, this relationship between physical activity and psychosocial well-being among adults with achondroplasia remains unexplored. Moreover, examinations of the gender differences in physical activity and psychosocial well-being within this population have been limited.

This study aimed to address these gaps in the literature by conducting an exploratory investigation of mental health symptoms, health-related quality of life, and physical activity levels among adults with achondroplasia, with a specific focus on examining potential gender differences in these relationships. By doing so, we aim to provide preliminary insights that can inform future research and interventions aimed at improving the overall well-being of adults with achondroplasia.

We hypothesized that higher levels of physical activity would be associated with better HrQoL and mental health outcomes in adults with achondroplasia, and that these relationships might differ by gender.

## 2. Materials and Methods

This cross-sectional, observational study examined relationships between physical activity, health-related quality of life, and mental health in adults with achondroplasia. The research design was chosen to provide a comprehensive snapshot of these variables in an understudied group of this rare population and to generate hypotheses for future longitudinal and interventional studies.

### 2.1. Participants

Sixteen Portuguese adults with achondroplasia, 10 women and 6 men aged between 20 and 57 years (37.2 ± 13.5), participated in this study. Participants were recruited through convenience sampling via the national association for skeletal dysplasia through direct contact and social media outreach. While specific prevalence data for achondroplasia in Portugal is not available, it is believed to align with global estimates of 1 in 25,000.

Inclusion criteria were: 1. Confirmed diagnosis of achondroplasia by genetic testing or clinical evaluation by a geneticist based on characteristic physical features and radiographic findings, and 2. age 18 years or older. No exclusion criteria were identified.

Anthropometric and body composition characteristics of participants are presented in Table 1. Sociodemographic characteristics and clinical history are presented in Table 2.

Notable findings included that 88% of participants did not have children, 44% lived with parents, 69% had a university degree, 63% were employed, 31% had undergone lower limb lengthening surgery, and 25% reported having sleep apnea. To minimize selection bias, the STROBE guidelines [15] for observational cross-sectional studies were applied to include a diverse sample in terms of gender, age, geographic location, and educational background. Additionally, the Sex and Gender Equity in Research guidelines (SAGER [16]) were used.

All participants received an informed consent document for review one week prior to data collection. Verbal explanations were provided to clarify any remaining doubts. A post-hoc power analysis was conducted using G*Power 3.1 [17] to determine the achieved power given our sample size. For detecting large effects (r = 0.7) with α = 0.05 in correlation analyses, our sample of 16 participants provided 80% power, suggesting adequate power for detecting substantial correlations, though likely underpowered for smaller effects.

This study was conducted in accordance with the Declaration of Helsinki and approved by the Ethics Committee of the (blinded for review), approval number 22052, on 6 June 2022. The study was conducted between November 2022 and March 2023, in (blinded for review).

### 2.2. Data Collection

Body weight, height, waist, hip circumference for waist/hip ratio (W/H), and body fat mass percentage (FM%) were collected for each participant as presented in Table 1. Body composition was assessed using a Tanita MC780-PMA bioelectrical impedance analyser (Tanita Corporation, Tokyo, Japan). Participants were measured wearing light clothing and barefooted, following standard protocols.

Participants completed a socio-demographic questionnaire which included household composition, parental status, education level, employment status, and clinical aspects, such as diagnosis of sleep apnea, frequent use of pain medication, and history of limb lengthening surgery. Participants completed three self-reporting validated questionnaires for the Portuguese population: the SF-36 version 2 (SF-36), the Brief Symptoms Inventory (BSI), and the International Physical Activity Questionnaire (IPAQ). Questionnaires were taken up to 7 days before physical assessments.

The SF-36 is a widely used instrument for assessing health-related quality of life across various populations, including those with chronic conditions [18]. It includes 36 questions across eight dimensions: physical functioning (PF), physical role (PR), bodily pain (Pain), general health perceptions (GH), vitality (VT), social functioning (SF), emotional role (ER), and general mental health (MH). Two summary measures, the physical component summary (PCS) and mental component summary (MCS), can also be calculated [19]. Scores for each dimension range from 0 to 100, with higher scores indicating better health status.

The BSI is a 53-item self-report inventory designed to assess psychological distress and psychiatric disorders. It measures nine primary symptom dimensions: somatization (Soma), obsessive-compulsive (ObsCom), interpersonal sensibility (IntSen), depression (Dep), anxiety (Anx), hostility (Hos), phobic anxiety (PhobA), paranoid ideation (ParaIdt), and psychoticism (Psyc) [20]. From results, three global indices can be calculated: the global severity index (GSI), positive symptom total (PST), and positive symptom distress index (PSDI). Higher scores indicate greater psychological distress. For the BSI, raw scores for the nine primary symptom dimensions range from 0 to 4, with higher scores indicating greater psychological distress. The three global indices (GSI, PST, and PSDI) are converted to standardized T-scores, with a mean of 50 and standard deviation of 10 in the normative sample. For these indices, T-scores ≥ 63 (93rd percentile) indicate clinically significant distress. Higher T-scores represent worse psychological health status. The GSI is the most sensitive indicator of the respondent’s distress level and combines information about the number of symptoms and the intensity of distress. The GSI is then transformed into the T-score GSI (TsGSI) [21]. Scores of 70 and above are considered representative of severe psychological distress [22], with higher scores representing worse psychological health status. The PTSI was particularly considered for this study, as in the reference population (RP) it presented a more significant F-statistic test, with scores above 1.7 indicative of emotionally perturbed cases [23].

The International Physical Activity Questionnaire (IPAQ) [24], short version, assesses physical activity levels over the past 7 days. It measures the frequency and duration of walking, moderate-intensity activities (M_MET), and vigorous-intensity activities (V_MET). IPAQ scores (PAS) are reported in metabolic equivalent (MET) minutes per week. A MET is a measure of energy expenditure, with one MET representing the energy expended while sitting quietly. IPAQ scores are categorized into three physical activity levels: low (<600 MET-min/week), moderate (600–3000 MET-min/week), or high (>3000 MET-min/week) physical activity levels. Higher MET-minutes/week indicate greater levels of physical activity. The data analyzed for this study are stored under a CC-By Attribution 4.0 International license at the Open Science Framework repository with the identifier https://doi.org/10.17605/OSF.IO/E4XNZ.

### 2.3. Data Analysis

Data analysis was conducted using SPSS software (IBM SPSS Statistics for Windows, IBM Corp, Armonk, NY, USA, version 29.0). Descriptive statistics were calculated for all anthropometric variables. The Shapiro–Wilk test was used to assess the normality of data distribution. Given the small sample size, non-normal distributed data, and subgroup analyses, nonparametric tests were employed. The Mann–Whitney U test was used to detect significant differences between subgroups. The effect size (r) for the Mann–Whitney U test was calculated as r = $Z/\sqrt{N}$, where Z is the standardized test statistic and N is the total sample size. Effect sizes were interpreted as small (r = 0.1), medium (r = 0.3), or large (r = 0.5). Fisher’s exact test was used to examine associations between categorical variables. Spearman’s correlation coefficient (rs) was calculated to assess relationships between quantitative variables [25]. The strength of correlations was interpreted as weak (0.1 ≤ |rs| < 0.3), moderate (0.3 ≤ |rs| < 0.5), or strong (|rs| ≥ 0.5). Linear regression analysis (R^2^) was also conducted to assess the proportion of variance in dependent variables explained by independent variables. To align the directionality of measures, BSI values were reverse-coded [26]. Cronbach’s alpha (α) was calculated to assess the internal consistency reliability of the SF-36 and BSI questionnaires, with values above 0.7 considered acceptable, above 0.8 considered good, and above 0.9 considered excellent. A *p*-value of <0.05 was considered statistically significant for all analyses. To address the increased risk of Type I errors due to multiple comparisons, we applied the Benjamini–Hochberg procedure to control the false discovery rate [27].

## 3. Results

### 3.1. Health-Related Quality of Life (SF-36)

SF-36 scores grouped by gender are presented in Table 3a. The individual score range of the SF-36 domains was particularly wide in physical function (10–100) and social function (12.5–100). The lowest mean score was found in pain (48.7 ± 13.8), while the highest was for physical role (71.90 ± 22.24). Women presented lower scores in all SF-36 domains except for pain (52.80 ± 15.99). The PCS was 46.7 ± 7.05 and the MCS was 45.1 ± 7.87. Significant differences between gender were observed in pain (U = 9.00, *p* = 0.021, r = 0.55) and in vitality (U = 7.00, *p* = 0.013, r = 0.61), with a strong effect size (r > 0.5).

### 3.2. Psychological Distress (BSI)

BSI results grouped by gender are presented in Table 3b. Notable findings are the lowest mean score observed in somatization (0.69 ± 0.78), with the highest score in paranoid ideation (1.24 ± 0.98). The individual score range was wider in anxiety (0.00–3.65) and obsession compulsion (0.33–3.67). It was noted that five adults (four women) presented a PSTI score above 2.0, indicating significant distress.

### 3.3. Physical Activity Levels (IPAQ)

IPAQ results grouped by gender are presented in Table 3c. Ten participants reported low physical activity level or being “inactive” (L1), while six reported being moderately or “minimally active” (L2). Significant gender difference in moderate-intensity activities (M_MET) were observed (U = 7.00, *p* = 0.007, r = 0.67), with a strong effect size (r > 0.5).

### 3.4. Associations between Physical Activity and Health Outcomes

When analyzing the SF-36 and BSI mean scores considering physical activity level grouping (results presented in Table 4a,b, respectively), significant differences were found in only two SF-36 dimensions, with physically active participants (L2) showing better outcomes in general health (U = 6.50, *p* = 0.012, r = 0.63), vitality (U = 9.50, *p* = 0.027, r = 0.55), and PCS (U = 10.0, *p* = 0.031, r = 0.67) compared to inactive participants (L1). However, after applying the Benjamini–Hochberg correction, these differences were not statistically significant, suggesting trends rather than definitive findings.

Notably, L1 participants presented lower scores in all SF-36 domains except for the emotional role (L1 = 65.0 ± 29.9 vs. L2 = 63.90 ± 24.0), as presented in Table 4a. Concerning the BSI questionnaire, L1 scored higher in all domains compared to L2 except for depression (L1 = 0.93 vs. L2 = 0.97). Interestingly, women and L1 participants presented the same PTSI (1.71) in both grouping situations.

Regarding associations between categorical variables, Fisher’s exact test revealed associations between gender and education (*p* = 0.036), with women more likely to have graduate studies, and between physical activity level and limb lengthening (*p* = 0.036), with adults with achondroplasia who had undergone lengthening likely to be more active.

The SF-36 and BSI demonstrated good to excellent internal consistency in this study sample, with Cronbach’s α of 0.838 for SF-36 and 0.960 for BSI.

### 3.5. Correlation and Regression Analysis

Correlation analysis revealed strong correlations (rs > 0.5) between body composition, physical activity measures, and SF-36 dimensions, as presented in Table 5. The strongest correlation was between general health and M_MET (rs = 0.786, *p* < 0.001). For negative correlations, several were found between FM% with physical activity measures and SF-36, with increased FM% more aligned with lesser physical fitness.

Linear regression analysis showed that moderate physical activity (M_MET) explained 72% of variance in general health scores (R^2^ = 0.720, *p* < 0.001), suggesting M_MET as a predictor of general health in this study sample. Additionally, rev_GSI also explained 58% of variance in mental component score (MCS) (R^2^ = 0.577) and 55% variance in social function (R^2^ = 0.551) and in mental health (R^2^ = 0.549), *p* < 0.001. The scatterplots shown in Figure 1 highlight a trend between moderate physical activity and general health (Figure 1a) and reversed-GSI and social function (Figure 1b).

## 4. Discussion

This exploratory study aimed to investigate relationships between physical activity, health-related quality of life, and mental health in adults with achondroplasia, with a focus on potential gender differences. The main findings partially support our initial hypothesis, providing preliminary insights into these relationships in this understudied population.

For the first hypothesis, that higher levels of physical activity would be associated with better HrQoL and mental health outcomes, we observed positive associations between moderate physical activity and several dimensions of health-related quality of life, particularly general health, vitality, and physical functioning. This aligns with the well-established benefits of physical activity in the general population [28]. Notably, moderate physical activity (M_MET) explained 72% of the variance in general health scores, suggesting a strong relationship between physical activity and perceived health status in adults with achondroplasia.

However, while we observed some trends suggesting better mental health outcomes in more physically active participants, these differences were not statistically significant after correcting for multiple comparisons. This ambiguity may be due to our small sample size or the complexity of factors influencing mental health in this population.

Our second hypothesis, that these relationships might differ by gender, was also partially supported. We observed significant gender differences in both physical activity levels and quality of life outcomes. Women reported lower scores in most SF-36 domains and were more likely to be physically inactive. These findings are consistent with previous research indicating that women with achondroplasia may face additional challenges related to self-esteem, body image, and social acceptance [12].

The study results suggest that adults with achondroplasia experience lower health-related quality of life and greater psychological distress compared to the general population, consistent with previous research in this area [13]. When comparing our results with normative data (RP) for the Portuguese population [28], adults with achondroplasia scored lower on most SF-36 dimensions, particularly in the pain domain, 30% below RP (48.69 ± 13.8 vs. 71.44 ± 24.27). This aligns with other previous findings of lower health-related quality of life in this population [4,6,8].

Interestingly, when analyzing data grouped by physical activity level, the moderately active subgroup (L2) scored higher than the reference population on several SF-36 dimensions including physical role, general health, and vitality. This finding suggests that physical activity may help mitigate quality of life deficits in adults with achondroplasia, though further research is needed to confirm this trend.

The BSI results indicated greater psychological distress in our sample compared to normative data from the Portuguese population, but lower distress compared to individuals with diagnosed emotional disturbances [23]. The largest differences from general population norms were seen for phobic anxiety (40% higher), psychoticism (32% higher), and anxiety (20% higher). The L1 subgroup exhibited the most severe distress, with scores indicating emotional disturbance. These results support previous reports of increased rates of anxiety and psychological disorders in adults with achondroplasia [7,12].

The exact mechanisms linking physical activity to improved quality of life and mental health in adults with achondroplasia remain to be elucidated. However, research in other populations suggests that physical activity promotes neuroplasticity, increases neurotrophic factors, and modulates neurotransmitter systems, all of which can contribute to improved mood and cognitive function [29,30]. Additionally, regular exercise can enhance self-efficacy, body image, and social interaction, which may be particularly beneficial for individuals with achondroplasia who often face unique psychosocial challenges [31]. Furthermore, physiological adaptations to exercise, such as improved cardiovascular fitness and musculoskeletal strength [32], may help mitigate some of the physical limitations associated with achondroplasia, thereby improving overall quality of life.

Our observation of gender differences in health outcomes and the identification of physically inactive individuals as a high-risk group for poor psychosocial outcomes have important implications for targeted physical activity interventions. These findings, even mostly being trends, suggest that physical activity promotion strategies for adults with achondroplasia may need to be tailored to address gender-specific challenges and to particularly engage those who are currently inactive.

Similar benefits of physical activity have been observed in populations with other conditions that affect mobility and stature. For instance, Shields et al. (2013) found that physical activity was associated with improved quality of life in adults with Down syndrome [33]. Additionally, Galhardo et al. (2024) reported positive effects of physical activity on autonomy, self-esteem, and fitness in individuals with osteogenesis imperfecta [34]. These findings suggest that the benefits of physical activity may extend across various conditions characterized by skeletal or mobility impairments.

It is also important to note that other interventions have also been explored to enhance quality of life in individuals with achondroplasia, mostly related to surgical interventions, such as limb lengthening, along with recently available pharmacologic treatments for children with achondroplasia [35]. Future research should explore how physical activity interventions might complement these existing approaches.

This study addresses important gaps in our understanding of the relationships between physical activity, quality of life, and mental health in adults with achondroplasia. By highlighting these associations and identifying key subgroups, our findings provide a foundation for future research aimed at enhancing the well-being of this understudied population.

### Strengths and Limitations

This study addresses a significant gap in the literature by examining relationships between physical activity, health-related quality of life, and mental health in adults with achondroplasia, using well-validated measures (SF-36, BSI, and IPAQ). The inclusion of both genders and a range of ages provides a more comprehensive view of the adult achondroplasia population than previous studies.

However, the small sample size (*n* = 16) limits statistical power and may mask smaller effect sizes, increasing the risk of Type II errors. While our post-hoc power analysis indicated adequate power for detecting large effects, the study’s cross-sectional design precludes causal inferences about the observed relationships. The reliance on self-reported physical activity may introduce recall bias and social desirability effects, suggesting future studies could benefit from incorporating objective measures such as accelerometers. Our recruitment through a national association for skeletal dysplasia, while facilitating access to this rare population, may have introduced selection bias, potentially limiting generalizability to the broader achondroplasia population.

The cross-sectional design precludes causal inferences about the relationships between physical activity and psychosocial outcomes. While multiple comparison corrections were performed, the study’s exploratory nature and small sample size suggest interpreting results as trends rather than definitive findings. Consequently, our results should be considered as generating hypotheses for future confirmatory studies rather than definitive findings. The absence of a matched control group limits our ability to determine whether the observed relationships are specific to adults with achondroplasia or reflect more general associations between physical activity and well-being. Despite these limitations, our study provides valuable preliminary data to inform future research and interventions in this understudied population, underscoring the need for larger, longitudinal studies with more diverse samples and robust methodologies to further elucidate these relationships in adults with achondroplasia.

## 5. Conclusions

This exploratory study provides preliminary evidence for the potential benefits of moderate-intensity physical activity on health-related quality of life and mental health among adults with achondroplasia.

Key findings include positive associations between moderate physical activity and dimensions of health-related quality of life, particularly general health and vitality, as well as notable gender differences in physical activity levels and quality of life outcomes.

These results have important implications for clinical practice and research. Healthcare providers should consider incorporating physical activity promotion, especially moderate-intensity activities, into care strategies for adults with achondroplasia. Particular attention should be given to developing targeted approaches for women and currently inactive individuals, who may be at higher risk for poor psychosocial outcomes.

Future research should focus on longitudinal and intervention studies with larger samples, investigating mechanisms linking physical activity to improved outcomes, and exploring integration with other therapeutic approaches. By addressing these research priorities, evidence-based strategies to enhance the overall well-being of adults with achondroplasia can be developed, ultimately improving their quality of life and long-term health outcomes.

## Figures and Tables

**Figure 1 ijerph-21-01160-f001:**
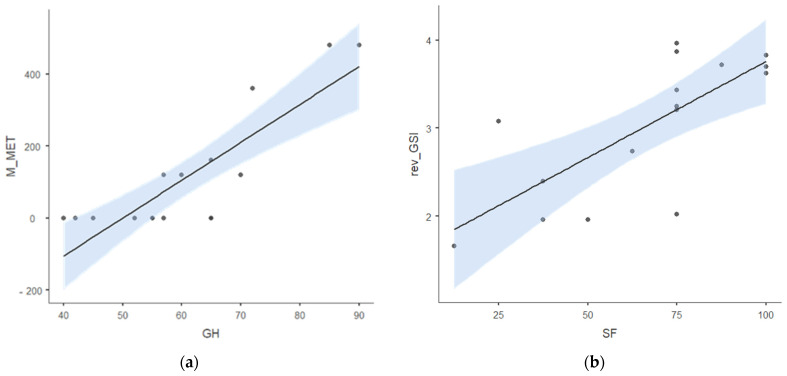
Linear regression between (**a**) general health and M_Met, (**b**) rev_GSI and SF. Standard error under blue area. Abbreviations: MET, metabolic equivalents; M_MET, moderate MET; GH, general health; SF, social functioning; rev_GSI, reverse-coding global severity index.

**Table 1 ijerph-21-01160-t001:** Anthropometric and body composition characteristics of participants, grouped by gender and total data. Data presented as mean ± SD.

	Age (Years)	Weight (kg)	Height (cm)	W/H	FM (%)
Women (*n* = 10)	34.8 ± 14.5	49.3 ± 10	123 ± 11.6	0.78 ± 0.04	27.8 ± 7.16
Men (*n* = 6)	41 ± 11.8	61.1 ± 18.5	130 ± 14.5	0.91 ± 0.05	26.5 ± 15.2
Total (*n* = 16)	37.1 ± 13.5	53.7 ± 14.5	125 ± 12.8	0.83 ± 0.08	27.3 ± 10.4

Abbreviations: W/H, waist to hip ratio; FM (%), fat mass percentage.

**Table 2 ijerph-21-01160-t002:** (**a**) Socio-demographic characteristics and (**b**) clinical history of study participants. Data presented as frequencies.

(**a**)			
**Sociodemographic Characteristic**		** *n* **	**Total%**
Household	Family of origin (FO)	7	43.8
	Out of FO	7	43.8
	Students (mix)	2	12.5
Children	No	14	87.5
	Yes	2	12.5
Education	Graduate	11	68.8
	Undergraduate	5	31.3
Work	Employed	10	62.5
	Student	4	25
	Unemployed	2	12.5
(**b**)			
**Clinical history**		** *n* **	**Total%**
Sleep Apnea	No	12	75
	Yes	4	25
Pain medication	No	13	81.25
	Yes	3	18.75
Limb lengthening	No	11	68.75
	Yes	5	31.25

**Table 3 ijerph-21-01160-t003:** (**a**) SF-36 and (**b**) BSI dimensions scores; (**c**) IPAQ data, grouped by gender and *p*-value for between group comparison.

(**a**)				
**SF-36**	**Women (*n* = 10)**	**Men (*n* = 6)**	**Total (*n* = 16)**	***p*-Value**
PF	60.00 ± 24.83	77.50 ± 28.06	66.6 ± 26.63	0.174
PR	65.00 ± 23.42	83.30 ± 15.65	71.90 ± 22.24	0.125
Pain	52.80 ± 15.99	41.80 ± 4.49	48.7 ± 13.79	0.021 *
GH	55.40 ± 10.25	70.20 ± 15.14	60.9 ± 13.93	0.081
Vital	44.40 ± 15.72	68.80 ± 9.68	53.5 ± 18.11	0.013 *
SF	57.50 ± 28.99	81.30 ± 15.31	66.4 ± 26.89	0.146
ER	59.20 ± 31.54	73.60 ± 15.54	64.6 ± 26.96	0.409
MH	61.50 ± 13.34	66.70 ± 8.76	63.4 ± 11.79	0.377
PCS	45.20 ± 6.89	49.20 ± 7.21	46.7 ± 7.05	0.220
MCS	42.50 ± 8.94	49.50 ± 2.47	45.1 ± 7.87	0.220
(**b**)				
**BSI**	**Women (*n* = 10)**	**Men (*n* = 6)**	**Total (*n* = 16)**	** *p* ** **-value**
Soma	0.79 ± 0.79	0.55 ± 0.82	0.69 ± 0.78	0.325
ObsCom	1.37 ± 1.04	1.00 ± 0.68	1.23 ± 0.92	0.511
IntSen	1.13 ± 1.00	0.58 ± 0.80	0.92 ± 0.94	0.264
Depres	1.08 ± 1.05	0.72 ± 0.75	0.95 ± 0.94	0.443
Anx	1.38 ± 1.12	0.83 ± 0.80	1.18 ± 1.02	0.414
Hos	0.82 ± 0.66	0.83 ± 0.71	0.82 ± 0.65	1.000
PhobA	0.84 ± 1.12	0.47 ± 0.72	0.7 ± 0.98	0.610
ParaIdt	1.30 ± 1.13	1.13 ± 0.74	1.24 ± 0.98	0.870
Psyc	1.10 ± 1.01	0.80 ± 0.99	0.99 ± 0.98	0.408
TsGSI	62.8 ± 9.92	59.00 ± 8.35	0.98 ± 0.79	0.480
GSI	1.28 ± 0.99	0.90 ± 0.84	59.75 ± 7.91	0.480
PST	29.20 ± 15.80	25.30 ± 17.10	27.75 ± 15.83	0.704
PSDI	1.71 ± 0.69	1.40 ± 0.39	1.59 ± 0.61	0.514
(**c**)				
**IPAQ**	**Women (*n* = 10)**	**Men (*n* = 6)**	**Total (*n* = 16)**	** *p* ** **-value**
V_MET	80 ± 155	480 ± 744	230 ± 480	0.260
M_MET	24 ± 50.6	267 ± 202	115 ± 173	0.007 *
W_MET	507 ± 518	435 ± 507	480 ± 498	0.624
PAS	599 ± 586	1181 ± 1071	817 ± 820	0.416
PAL ^†^ *	1	2	1	

Abbreviations: PF, physical function; PR, physical role; Pain, bodily pain; GH, general health perceptions; Vital, vitality; SF, social function; ER, emotional role; MH, mental health; PCS, physical component score; MCS, mental component score; Soma, somatization; ObsCom, obsessive-compulsive; IntSen, interpersonal sensibility; Depres, depression; Anx, anxiety; Hos, hostility; PhobA, phobic anxiety; ParaIdt, paranoid ideation; Psyc, psychoticism; TsGSI, T-score global severity index; GSI, global severity index; PST, positive symptom total; PSDI, positive symptom distress index. MET, metabolic equivalents; W_MET, walking MET; M_MET, moderate MET; V_MET, vigorous MET; PAS, physical activity score; PAL, Physical activity level. ^†^ Data presented as median. *, significant *p*-value for groups difference.

**Table 4 ijerph-21-01160-t004:** Between-groups comparisons (L1 and L2), with level of significance for (**a**) Short Form-36 (SF-36) and (**b**) BSI questionnaire dimensions.

(**a**)				
**SF-36**	**L1 (*n* = 10)**	**L2 (*n* = 6)**	***p*-Value**	***p*-Value Corr**
PF	59.0 ± 27.9	79.2 ± 20.6	0.158	0.316
PR	66.9 ± 23.2	80.2 ± 19.5	0.273	0.455
Pain	47.2 ± 14.1	51.2 ± 14.2	1.000	1.000
GH	54.1 ± 9.71	72.3 ± 12.8	0.012 *	0.120
Vital	45.6 ± 16.2	66.7 ± 13.5	0.027 *	0.135
SF	58.8 ± 27.0	79.2 ± 23.3	0.118	0.295
ER	65.0 ± 29.9	63.9 ± 24.0	0.912	1.000
MH	62.0 ± 10.3	65.8 ± 14.6	0.581	0.831
PCS	43.8 ± 6.21	51.6 ± 5.81	0.031 *	0.103
MCS	44.3 ± 8.0	46.5 ± 8.21	0.635	0.794
(**b**)				
**BSI**	**L1 (*n* = 10)**	**L2 (*n* = 6)**	** *p* ** **-value**	** *p* ** **-value corr**
Soma	0.73 ± 0.82	0.64 ± 0.78	0.956	1.000
ObsCom	1.45 ± 1.03	0.86 ± 0.59	0.208	1.000
IntSen	1.05 ± 0.87	0.78 ± 1.10	0.288	1.000
Depres	0.93 ± 0.86	0.97 ± 1.15	0.913	1.000
Anx	1.33 ± 1.09	0.92 ± 0.94	0.414	0.597
Hos	0.92 ± 0.63	0.67 ± 0.72	0.350	0.650
PhobA	0.88 ± 1.11	0.4 ± 0.70	0.395	0.642
ParaIdt	1.48 ± 1.07	0.83 ± 0.71	0.276	1.000
Psyc	1.10 ± 0.90	0.8 ± 1.17	0.508	0.660
TsGSI	60.90 ± 8.06	57.8 ± 7.97	0.356	0.926
GSI	1.09 ± 0.81	0.78 ± 0.79	0.356	0.771
PST	29.7 ± 15.0	24.5 ± 18.10	0.625	0.741
PSDI	1.71 ± 0.66	1.39 ± 0.49	0.355	1.000

Abbreviations: L1, physical activity level 1; L2, physical activity level 2; PF, physical functioning; PR, physical role; Pain, bodily pain; GH, general health perceptions; Vital, vitality; SF, social functioning; ER, emotional role; MH, mental health; PCS, physical component score; MCS, mental component score; Soma, somatization; ObsCom, obsessive-compulsive; IntSen, interpersonal sensibility; Depres, depression; Anx, anxiety; Hos, hostility; PhobA, phobic anxiety; ParaIdt, paranoid ideation; Psyc, psychoticism; TsGSI, T-score global severity index; GSI, global severity index; PST, positive symptom total; PSDI, positive symptom distress index. Results are presented as mean and standard deviations. *, significant difference between PAL groups; *p*-value corr, corrected *p*-values using Benjamini–Hochberg procedure.

**Table 5 ijerph-21-01160-t005:** Significant (*p* < 0.05) Spearman correlations coefficients, presented as rs. Other levels of significance are presented as * *p* < 0.01, ** *p* < 0.001.

	FM%	V_MET	M_MET	PAS	PF	PR	GH	Vital	SF	MH	MCS
**V_MET**	−0.53										
**M_MET**	−0.53										
**W_MET**	−0.61										
**PAS**	−0.83 **										
**PF**	−0.58		0.62 *	0.64 *							
**Pain**			−0.57								
**GH**	−0.59		0.79 **	0.61							
**Vital**			0.60								
**PCS**	−0.60	0.54	0.59	0.72 *							
**rev_Soma**								0.67 *	0.58	0.75 **	0.75 **
**rev_ObComp**							0.65 *	0.71 *	0.75 **	0.76 **	0.72 *
**rev_IntSen**							0.77 **	0.77 **	0.69 *	0.78 **	0.81 **
**rev_Dep**							0.66 *	0.57	0.63 *	0.78 **	0.69 *
**rev_Anx**								0.70 *	0.69 *	0.78 **	0.78 **
**rev_Hos**							0.56	0.59	0.66 *	0.67 *	0.71 *
**rev_PhobA**					0.61	0.56	0.52	0.60	0.72 *	0.65 *	0.63 *
**rev_ParaIdt**							0.65 *	0.49	0.65 *	0.64 *	0.65 *
**rev_Psyc**							0.60	0.63 *	0.81 **	0.69 *	0.81 **
**rev_GSI**							0.64 *	0.71 *	0.75 **	0.81 **	0.80 **
**rev_TsGSI**							0.64 *	0.71 *	0.75 **	0.81 **	0.80 **

Abbreviations: MET, metabolic equivalents; W_MET, walking MET; M_MET, moderate MET; V_MET, vigorous MET; PAS, physical activity score; PF, physical functioning; PR, physical role; Pain, bodily pain; GH, general health perceptions; Vital, vitality; SF, social functioning,; MH, mental health; PCS, physical component score; rev, reverse-coding; Soma, somatization; ObsCom, obsessive-compulsive; IntSen, interpersonal sensibility; Dep, depression; Anx, anxiety; Hos, hostility; PhobA, phobic anxiety; ParaIdt, paranoid ideation; Psyc, psychoticism; TsGSI, T-score global severity index; GSI, global severity index.

## Data Availability

Data supporting the reported results can be found at the repository Open Science Framework, at https://osf.io/e4xnz/ (accessed on 1 August 2024) with the identifier https://doi.org/10.17605/OSF.IO/E4XNZ.

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
