# Peer review of "Physical Activity and Psychosocial Outcomes in Adults with Achondroplasia: An Exploratory Study"

_ijerph, 2024, doi:10.3390/ijerph21091160_

Round 1

Reviewer 1 Report

Comments and Suggestions for Authors

The manuscript that's been reviewed is ( PA &Qol& MH in adults with achondroplesia) Intersting study that include people with rare condition.

Please check the following comments :

Abstract : Need to be more inclusive of the article please clearlg write the aim os study based on strong background .Remove abbreviation from abstrat end it with your findings only without needs for further research.

Your sample size is only 16 which is too samll sample size for such cross sectional study so you have to add this ad limitation that affect generlization of study. Actually you have shift type of study to pilot study 

Methods : As you werd working on portogeses adults how you srlected the sample size? Pleasr write prevelance of achondroplesia in Portuguese. 

In discussion add literature studying effect of PA on population with other similar conditions .Also discuss other interventions used with achondroplesia to enhance Qol.

In conclusion need to clearly articulate recommendations for further research.

Author Response

COMMENTS

  1. Abstract: Need to be more inclusive of the article please clearly write the aim of study based on strong background. Remove abbreviation from abstract end it with your findings only without needs for further research.

Thank you for this feedback. We have revised the abstract to be more comprehensive and clearly state the study aim based on a strong background. We have removed abbreviations and focus on presenting the key findings without mentioning the need for further research. Changes made in lines 13-30.

  1. Your sample size is only 16 which is too small sample size for such cross-sectional study, so you must add this limitation that affect generalization of study. You have shift type of study to pilot study.

We acknowledge this limitation. In lines 359-360, we have added: " However, the small sample size (n=16) limits statistical power and may mask smaller effect sizes, increasing the risk of Type II errors. While our post-hoc power analysis indicated adequate power for detecting large effects, the study's cross-sectional design precludes causal inferences about the observed relationships.

In fact, we have conducted a pilot study with only 4 adults in 2022, and this cohort was the larger sample. Yet, as this research relates to a rare population and a subgroup (adults), with increased difficulties to involve in research, this was the possible population. 

  1. Methods: As you were working on Portuguese’s adults how you selected the sample size? Please write prevalence of achondroplasia in Portuguese.

We appreciate raising this point. In lines 94-96, we have clarified this point: “Participants were recruited through convenience sampling via the national association for skeletal dysplasia through direct contact and social media outreach. While specific prevalence data for achondroplasia in Portugal is not available, it is believed to align with global estimates of 1:25,000.”

  1. In discussion add literature studying effect of PA on population with other similar conditions.

We thank the reviewer for this suggestion. We have added the following paragraph (lines 335-341):

Benefits of physical activity have been observed in populations with other conditions that affect mobility and stature. For instance, Shields et al. (2013) found that physical activity was associated with improved quality of life in adults with Down syndrome[34]. Additionally, Galhardo et al. (2024) reported positive effects of physical activity on autonomy, self-esteem, and fitness in individuals with osteogenesis imperfecta[35].These findings suggest that the benefits of physical activity may extend across various conditions characterized by skeletal or mobility impairments.

  1. Also, discuss other interventions used with achondroplasia to enhance Qol.

We have added the following information (lines 342-346):

“It's also important to note that other interventions have also been explored to enhance quality of life in individuals with achondroplasia, mostly related to surgical interventions such as limb lengthening and recently available pharmacologic treatments for children with achondroplasia [36]. Future research should explore how physical activity interventions might complement these existing approaches.”

  1. In conclusion need to clearly articulate recommendations for further research.

We have revised the conclusion to more clearly articulate recommendations for further research (393-397):

“Future research should focus on longitudinal and intervention studies with larger samples, investigating mechanisms linking physical activity to improved outcomes, and exploring integration with other therapeutic approaches. By addressing these research priorities, evidence-based strategies to enhance the overall well-being of adults with achondroplasia can be developed to, ultimately improving their quality of life and long-term health outcomes.”

Reviewer 2 Report

Comments and Suggestions for Authors

This exploratory, cross-sectional study explored relationships between health-related quality of life, mental health, and physical activity levels in this population, in adults with achondroplasia. The study’s premise is relevant and significant. However, certain points require further clarification and attention. These are as follows:

1.     Pg 1: abstract. To provide concrete evidence to support claims made in the abstract, the results should include quantitative statistical measures (such as p-values, correlation coefficient, confidence intervals of the correlation …. etc).

2.     Pg 2; Ln 68-70. You pinpointed a gap in the literature that “gender differences in physical activity and psychosocial well-being within this population have not been adequately examined in adults with achondroplasia”, however, the study's objective doesn’t explicitly state whether this gap is addressed within the scope of the study.

3.     The sample size appears insufficient to reliably detect the relationships of interest. With only 16 participants, the statistical power may be inadequate to draw a definitive conclusion about the relationships under investigation.

4.     Pg 2-3; Ln 96-99. Please provide proper credit to the source of the information employed for the sample size calculation.

5.     Pg 3; Ln 112-140. How each of these measures (i.e., SF-36, BSI, and IPAQ) was scored and how could these scores be interpreted is worthy of mention.

6.     Tables 3 and 4, which present the sample performance data for the SF-36, BSI, and IPAQ can be merged into a single table.

7.     Table 5; could you drop a column demonstrating the significance level for the between-group comparison? It would be interesting if you could report calculated p and corrected p values.

8.     Pg 7; Ln 239-242; Table 6. I can see results about the internal consistency of SF-36 items. Are these results within the scope of the study? Are they important to mention? If yes, justify and refer to the test in the statistical analysis subsection of the methods.

9.     Spearman’s correlation coefficient (rs) was calculated to assess relationships between quantitative variables, but I could not find the results of this analysis.

10.  The interpretation of the linear regression analysis is inaccurate. The R-squared value refers to the proportion of variation in the dependent variable explained by the independent variable, not the strength of correlation. The p-value indicates that the coefficient is significantly different from zero, suggesting that the independent variable has a meaningful impact on the dependent variable.

11.  I would suggest revising and reorganizing the results section to help readers follow the study results based on the objectives stated in the introduction. Also, the discussion section should be revised accordingly.

12.  Pg 10; Ln 313-316. It is not necessary to repeat descriptive statistics in the discussion. This section should focus on the interpretation and implication of the results, rather than repeating them.

13.  Pg 11; Ln 373-393. Please, revise the conclusion for briefness and conciseness. Summarize the key findings, articulate the study's significance, and briefly outline implications for clinical practice and future research directions.

Author Response

COMMENTS

This exploratory, cross-sectional study explored relationships between health-related quality of life, mental health, and physical activity levels in this population, in adults with achondroplasia. The study’s premise is relevant and significant. However, certain points require further clarification and attention. These are as follows:

We appreciate the thorough review and valuable suggestions provided. These changes will significantly improve the clarity, rigor, and impact of our manuscript.

  1. Pg 1: abstract. To provide concrete evidence to support claims made in the abstract, the results should include quantitative statistical measures (such as p-values, correlation coefficient, confidence intervals of the correlation …. etc).

We appreciate this suggestion. We have revised the abstract (lines 13-30) to include key statistical measures. “Moderate physical activity showed strong positive correlations with general health (rs=0.79, 95% CI [0.50,0.92], vitality (rs=0.60, 95% CI [0.15,0.85], and physical functioning (rs=0.62,95% CI [0.18,0.86], on SF-36.”

  1. Pg 2; Ln 68-70. You pinpointed a gap in the literature that “gender differences in physical activity and psychosocial well-being within this population have not been adequately examined in adults with achondroplasia”, however, the study's objective doesn’t explicitly state whether this gap is addressed within the scope of the study.

Thank you for pointing this out. We have revised the text to clearly present the study objectives (lines 74-77): " This study aimed to address these gaps in the literature by conducting an exploratory investigation of mental health symptoms, health-related quality of life, and physical activity levels among adults with achondroplasia, with a specific focus on examining potential gender differences in these relationships”

  1. The sample size appears insufficient to reliably detect the relationships of interest. With only 16 participants, the statistical power may be inadequate to draw a definitive conclusion about the relationships under investigation.

We acknowledge this limitation. In lines 359-360, we have added: " However, the small sample size (n=16) limits statistical power and may mask smaller effect sizes, increasing the risk of Type II errors. While our post-hoc power analysis indicated adequate power for detecting large effects, the study's cross-sectional design precludes causal inferences about the observed relationships.

  1. Pg 2-3; Ln 96-99. Please provide proper credit to the source of the information employed for the sample size calculation.

We apologize for the oversight. We have reconsidered this information and acknowledge that a post-hoc analysis is a more adequate information, presented now in lines 110-114: "A post-hoc power analysis was conducted using G*Power 3.1 (Faul et al., 2007) [17] to determine the achieved power given our sample size. For detecting large effects (r = 0.7) with α = 0.05 in correlation analyses, our sample of 16 participants provided 80% power, suggesting adequate power for detecting substantial correlations, though likely underpowered for smaller effects."

  1. Pg 3; Ln 112-140. How each of these measures (i.e., SF-36, BSI, and IPAQ) was scored and how could these scores be interpreted is worthy of mention.

We have added information for each questionnaire:

For the SF-36 (lines144-145): “Scores for each domain range from 0-100, with higher scores indicating better health status.

For the BSI (line152-157): “For the BSI, raw scores for the nine primary symptom dimensions range from 0-4, with higher scores indicating greater psychological distress. The three global indices (GSI, PST, and PSDI) are converted to standardized T-scores, with a mean of 50 and standard deviation of 10 in the normative sample. For these indices, T-scores ≥63 (93rd percentile) indicate clinically significant distress. Higher T-scores represent worse psychological health status”

For the IPAQ (lines 168-173): “IPAQ scores (PAS) are reported in metabolic equivalent (MET) minutes per week. A MET is a measure of energy expenditure, with one MET representing the energy expended while sitting quietly. IPAQ scores are categorized into three physical activity levels: low (<600 MET-min/week), moderate (600-3000 MET-min/week), or high (>3000 MET-min/week) physical activity levels. Higher MET-minutes/week indicate greater levels of physical activity.”

  1. Tables 3 and 4, which present the sample performance data for the SF-36, BSI, and IPAQ can be merged into a single table.

We appreciate this suggestion, and we have now merged Tables 3 and 4 in the revised manuscript, which is now Table 3 with a, b, c sections, and can be found in page 5, starting at line 218. We have also added a column with group difference significance (with Mann-Whitney test p-value).

  1. Table 5; could you drop a column demonstrating the significance level for the between-group comparison? It would be interesting if you could report calculated p and corrected p values.

Table 5 has been renamed to Table 4 (starting at line 242). We have added two columns, one with Mann-Whitney test p-value and “p-value corr”, showing Benjamini-Hochberg corrected p-values for between-group comparisons. We have verified the calculations, and no significance remained after correction. Therefore, we have corrected the following text in the results (lines 234-236): “Yet, after applying the Benjamini-Hochberg correction, these differences were not statistically significant, suggesting trends rather than definitive findings.” We have also included in the Limitations section (lines 377-379): “Yet, after applying the Benjamini-Hochberg correction, these differences were not statistically significant, suggesting trends rather than definitive findings.”

  1. Pg 7; Ln 239-242; Table 6. I can see results about the internal consistency of SF-36 items. Are these results within the scope of the study? Are they important to mention? If yes, justify and refer to the test in the statistical analysis subsection of the methods.

We agree that this information is not central to the study's main objectives. We will remove Table 6 and the associated text. We have added in the Methods, 2.3 Data analysis the following “Cronbach's alpha (α) was calculated to assess the internal consistency reliability of the SF-36 and BSI questionnaires, with values above 0.7 considered acceptable, above 0.8 considered good, and above 0.9 considered excellent.”

We have added also in Results (lines 253-254): “The SF-36 and BSI demonstrated good to excellent internal consistency in this study sample, with Cronbach's α of 0.838 for SF-36 and 0.960 for BSI”.

  1. Spearman’s correlation coefficient (rs) was calculated to assess relationships between quantitative variables, but I could not find the results of this analysis.

We apologize for the confusion. The Spearman correlation results are indeed presented in Table 5 (previously table 7), starting from line 261. This table provides a comprehensive matrix of significant Spearman's correlation coefficients (rs) between various measures, including physical activity variables, SF-36 domains, and BSI dimensions. The table includes significance levels indicated by asterisks. As only significant (p<0.05) correlations have been included in the table, we chose to mark with* p<0.01 and ** p<0.001, for readability. We have ensured that the reference to Table 5 is clear in the text when discussing correlation results (line 257).

  1. The interpretation of the linear regression analysis is inaccurate. The R-squared value refers to the proportion of variation in the dependent variable explained by the independent variable, not the strength of correlation. The p-value indicates that the coefficient is significantly different from zero, suggesting that the independent variable has a meaningful impact on the dependent variable.

Thank you for this correction. We have revised the interpretation (lines 271-275) to: "Linear regression analysis showed that moderate physical activity (M_MET) explained 72% of the variance in General Health scores (R²=0.720, p<0.001), suggesting that M_MET is a significant predictor of General Health in this sample. Additionally, rev_GSI also explained 58% of the variance in Mental component score (R²=0.577) and 55% in Social Function (R²=0.551) and in Mental Health (R²=0.549), p<0.001.”

  1. I would suggest revising and reorganizing the results section to help readers follow the study results based on the objectives stated in the introduction. Also, the discussion section should be revised accordingly.

We appreciate this suggestion, and we have reorganize the Results and Discussion sections to be organized and aligned with the study objectives stated in the Introduction.

  1. Pg 10; Ln 313-316. It is not necessary to repeat descriptive statistics in the discussion. This section should focus on the interpretation and implication of the results, rather than repeating them.

Thank you for this remark. Nevertheless, the descriptive data presented at the Discussion is related to the comparison with normative data for the Portuguese population, offering a more comprehensive framework of the results. We consider a valorous input to present this data in the discussion as these is not our study data, yet if the reviewer still consider that this is not relevant to be presented, we will remove the information of results comparison with normative population data.

We have reworded the initial paragraph (lines 310-314): “When comparing our results with normative data (RP) for the Portuguese population [30], adults with achondroplasia scored lower on most SF-36 dimensions, particularly in the Pain domain, 30% below RP (48.69±13.8 vs 71.44±24.27). This aligns with other previous findings of lower health-related quality of life in this population [4, 6, 8].”

  1. Pg 11; Ln 373-393. Please, revise the conclusion for briefness and conciseness. Summarize the key findings, articulate the study's significance, and briefly outline implications for clinical practice and future research directions.

Thank you for your suggestion. We have revised the conclusion (lines 389-405) for brevity and conciseness, focusing on key findings, significance, and implications for practice and research.

Reviewer 3 Report

Comments and Suggestions for Authors

Your research on the extremely rare condition of achondroplasia is highly significant and intriguing. However, there are some issues with your manuscript, particularly in the discussion and conclusion sections.

Firstly, in the discussion, I feel that the results of your study have been overly generalized. Specifically, there is a lack of logical basis for applying the obtained results to other populations or conditions, leading to a leap in the conclusions. The discussion should be limited to a more restrained interpretation of the results, avoiding excessive speculation and claims. Additionally, there is insufficient alignment with existing literature, making it unclear how the findings of this study should be positioned.

Secondly, the practical implications and applicability suggested in the conclusion are also questionable due to the lack of sufficient evidence at this stage. These claims should be reconsidered.

Author Response

COMMENTS

Your research on the extremely rare condition of achondroplasia is highly significant and intriguing. However, there are some issues with your manuscript, particularly in the discussion and conclusion sections.

1.In the discussion, I feel that the results of your study have been overly generalized. Specifically, there is a lack of logical basis for applying the obtained results to other populations or conditions, leading to a leap in the conclusions. The discussion should be limited to a more restrained interpretation of the results, avoiding excessive speculation and claims. Additionally, there is insufficient alignment with existing literature, making it unclear how the findings of this study should be positioned.

We appreciate this feedback. We have deeply revised the discussion, considering an alignment with the research questions, and with a restrained interpretation of our results. We have added the following paragraph to the beginning of the Discussion section (line 285 and after):

This exploratory study aimed to investigate relationships between physical activity, health-related quality of life, and mental health in adults with achondroplasia, with a focus on potential gender differences. The main findings partially support our initial hypothesis, providing preliminary insights into these relationships in this understudied population.”

Lines 339-340 “These findings, even mostly being trends…”

We will also revise the discussion to better align with existing literature by adding the following paragraph (lines 343-349):

Similar benefits of physical activity have been observed in populations with other conditions that affect mobility and stature. For instance, Shields et al. (2013) found that physical activity was associated with improved quality of life in adults with Down syndrome [33]. Additionally, Galhardo et al. (2024) reported positive effects of physical activity on autonomy, self-esteem, and fitness in individuals with osteogenesis imperfecta [34]. These findings suggest that the benefits of physical activity may extend across various conditions characterized by skeletal or mobility impairments. These findings suggest that the benefits of physical activity may extend across various conditions characterized by skeletal or mobility impairments.”

  1. The practical implications and applicability suggested in the conclusion are also questionable due to the lack of sufficient evidence at this stage. These claims should be reconsidered.

We agree that our claims regarding practical implications should be more cautious. We have highlighted several limitations of our study under the limitation section and have also revised the conclusion section for conciseness and clarity. It can now be read between lines 389-394 the following:

This exploratory study provides preliminary evidence for the potential benefits of moderate-intensity physical activity on health-related quality of life and mental health among adults with achondroplasia. Key findings include positive associations between moderate physical activity and dimensions of health-related quality of life, particularly general health and vitality, as well as notable gender differences in physical activity levels and quality of life outcomes.”

Reviewer 4 Report

Comments and Suggestions for Authors

The authors must be commended for carrying out a study on the relationship between physical activity level, quality of life and mental health in individuals with achondroplasia. This topic is very interesting and important. The research methodology used in the study is appropriate, and the manuscript is written with good clarity. However, some issues need to be taken into consideration.

Abstract

Please add some quantitative indicators of observed results (e.g. p<0.05).

Introduction

Please divide the introduction section into paragraphs.

Add a hypothesis at the end of the introduction section.

Methods

Please emphasize the study design in the methods section.

Participants: Emphasize a place in the text where the reader can see information about participants (Table 1).

Please add a criteria for achondroplasia diagnosis.  

Data collection: Please add information about bioimpedance manufacturers. Also, add some (brief) information about the procedure for collecting body composition data.

Results

I suggest moving Table 1 to the Methods section.

3.4.: Please emphasize a place in the results where the reader can see this information (Table 4).

3.5.: Please emphasize a place in the results where the reader can see this information. Also, emphasize in the Table the observed differences.

Table 5: Emphasize in the Table the observed differences.

Table 7: It is unclear to me, what is the asterisk for the p < 0.05?

Discussion

I suggest starting the discussion section with the main findings of your study.

When referring to the study results, please emphasize the place in the results where the reader can see them (Table 4, e.g.).

Conclusion

Remove the reference from the conclusion section.

I strongly suggest shortening the conclusion section, focusing on the most important finding of your study.

Author Response

COMMENTS

The authors must be commended for carrying out a study on the relationship between physical activity level, quality of life and mental health in individuals with achondroplasia. This topic is very interesting and important. The research methodology used in the study is appropriate, and the manuscript is written with good clarity. However, some issues need to be taken into consideration.

Abstract

  1. Please add some quantitative indicators of observed results (e.g. p<0.05).

 We have added quantitative indicators to the abstract (page 1, lines 20-25) as follows:

Moderate physical activity showed strong positive correlations with general health (rs=0.79, 95% CI [0.50,0.92], vitality (rs=0.60, 95% CI [0.15,0.85], and physical functioning (rs=0.62,95% CI [0.18,0.86], on SF-36. Women reported lower quality of life scores than men across most SF-36 dimensions. Significant gender difference was observed in Vitality (r=0.61) and Pain (r=0.55). Physically active participants presented better outcomes in General health (r=0.63) and Vitality (r=0.55) compared to inactive participants.

Introduction

  1. Please divide the introduction section into paragraphs.

Thank you for this suggestion. We have divided the introduction into paragraphs for improved readability, flow and clarity.

  1. Add a hypothesis at the end of the introduction section.

We also have added a hypothesis at the end of the introduction (lines 80-82): "We hypothesized that higher levels of physical activity would be associated with better HrQoL and mental health outcomes in adults with achondroplasia, and that these relationships might differ by gender." 

Methods

  1. Please emphasize the study design in the methods section.

We have emphasized the study design in the first paragraph of the Methods section (line 86-90): "This cross-sectional, observational study examined relationships between physical activity, health-related quality of life, and mental health in adults with achondroplasia. The research design was chosen to provide a comprehensive snapshot of these variables in an understudied group of this rare population and to generate hypotheses for future longitudinal and interventional studies

  1. Participants: Emphasize a place in the text where the reader can see information about participants (Table 1).

We have added a reference to Table 1 in the Participants section (lines 101-102): "Anthropometric and body composition characteristics of participants are presented in Table 1. Sociodemographic characteristics and clinical history are presented in Table 2.

  1. Please add a criteria for achondroplasia diagnosis.  

We have added criteria for achondroplasia diagnosis (98-100): “Inclusion criteria were: 1. Confirmed diagnosis of achondroplasia by genetic testing or clinical evaluation by a geneticist based on characteristic physical features and radiographic findings…”

  1. Data collection: Please add information about bioimpedance manufacturers. Also, add some (brief) information about the procedure for collecting body composition data.

We have added information about bioimpedance and body composition data collection (lines 121-123): "Body composition was assessed using a Tanita MC780-PMA bioelectrical impedance analyser (Tanita Corporation, Tokyo, Japan). Participants were measured wearing light clothing and barefooted, following standard protocols."

Results

  1. I suggest moving Table 1 to the Methods section.

We acknowledge the clarity of this change. We have move Table 1 to section 2.2 Data collection, under Materials and methods section.

  1. Section 3.4.: Please emphasize a place in the results where the reader can see this information (Table 4).

Table 4 has been renamed as Table 3, which includes 3 subsections now: 3a. SF-36, 3b. BSI and 3c. IPAQ. This information is presented in lines 199, 207 and 213, respectively.

  1. Section 3.5.: Please emphasize a place in the results where the reader can see this information. Also, emphasize in the Table the observed differences.

Thank you for identifying this need for clarity. We have removed the section “Gender differences” and embed the information related to this topic in Table 3 and in sections 3.2 and 3.4.  We have also added in the Table 3 the p-value of significance level of gender differences.

  1. Table 5: Emphasize in the Table the observed differences.

Table 5 has been renamed Table 4. We have added asterisks for significant differences between groups.

  1. Table 7: It is unclear to me, what is the asterisk for the p < 0.05?

Thank you for this remark. We have renamed this as Table5. We have included only significant correlations, therefore only correlation with p<0.05 were presented. To reduced space for the table, we have chosen to mark with asterisks only those significant correlations with p<0.01 (*) and p<0.001 (**). We have clarified this option in the title of Table 5 (lines 261-262) Table 5. Significant (p<0.05) Spearman correlations coefficients, presented as rs. Other levels of significance are presented as *p < 0.01, **p < 0.001”

Discussion

  1. I suggest starting the discussion section with the main findings of your study.

We have updated the beginning of the Discussion section, by adding in the first paragraph the main findings (line 287-290):

"The main findings of this exploratory study suggest that moderate physical activity is positively associated with several dimensions of health-related quality of life in adults with achondroplasia, particularly general health, vitality, and physical functioning. Additionally, we observed gender differences in both physical activity levels and quality of life outcomes."

  1. When referring to the study results, please emphasize the place in the results where the reader can see them (Table 4, e.g.).

We have searched for missing references and added some about the specific tables in the results. Lines 199, 207, 213, 231, 257, 276

Conclusion

  1. Remove the reference from the conclusion section.

References have been removed from the conclusion section.

  1. I strongly suggest shortening the conclusion section, focusing on the most important finding of your study.

Thank you for this suggestion. We have shortened the conclusion to focus on the most important findings (lines 389-405), and future directions of research.

Round 2

Reviewer 1 Report

Comments and Suggestions for Authors

Required modifications were done but still need to add pilot study to the title 

Author Response

Thank you for your comment. After the 1st round of review, we conducted a profound revision of the manuscript, in which the small sample. was presented as one of the limitations of the study. This is mostly due achondroplasia being a rare condition and our work being carried out in the adult's subgroup of this populations. We acknowledge your indication to present this work as a pilot study, yet we have conducted a pilot study, in early 2022, which included only 4 adults. Considering that along the manuscript text, the study is presented as "exploratory" we have revised the title to "Physical Activity and Psychosocial Outcomes in Adults with Achondroplasia: An Exploratory Study". As exploratory instead of pilot, we consider that it is clearer that the relation between physical activity and psychosocial outcomes have not been investigated and its connection are unclear. We look forward in receiving your feedback on this matter. 

Reviewer 2 Report

Comments and Suggestions for Authors

I acknowledge the diligent effort the authors have made in addressing the comments and suggestions provided in the previous version. The manuscript demonstrated significant improvement in terms of clarity and overall presentation. In my point of view, the manuscript is publishable in its current form.

Author Response

Thank you very much for your feedback to the correction done to the manuscript. This process was very intense, and your reflections and suggestions were extremely important to improve this manuscript. We truly appreciate your contributions. We have solely updated the title in this second round, following another reviewer comments to "Physical Activity and Psychosocial Outcomes in Adults with Achondroplasia: An Exploratory Study", as "exploratory" better presentes the lack of investigation in this area.